# Research on Platform Operation Strategy Considering Consumers' Hassle Costs

**Huixin Liu \*, Fang Liu and Feng Du**

School of Management Engineering, Zhengzhou University, Zhengzhou 450001, China;
l18749123649@163.com (F.L.); 18839100307@163.com (F.D.)

\* Correspondence: liuhuixin@zzu.edu.cn

**Abstract:** Based on consumers' different preferences for hassle costs, we study two platform operation strategies: selected platforms and diversified platforms. Considering diverse charging systems of merchants on the platform, a two-sided user utility function and profit function are established to examine the influence of hassle costs, platform services and the strength of two-sided network effects on the scale of platform users, and platforms' profits and price. The results show that: (1) The selected platform strategy adopting the transaction fee system is better than other strategies. (2) Under the selected platform strategy, the scale of bilateral users and platform profits will decrease with the increase in hassle costs, and increase with the strengthening of the bilateral network effects. However, the proportion of equilibrium pricing for merchants will increase with the increase in consumer hassle costs, and will decrease with the increase in the network effect on the consumer side. (3) The less value-added services that selected platforms provide to consumers, the more value-added services exist to merchants and the higher the equilibrium pricing is for merchants. However, as the network effect on the side of merchants is increasing, the equilibrium pricing ratio of the platform to merchants shows three trends However, the general trend is that the greater the network effect of the business side, the lower the fee ratio and the higher the platform profit.

**Keywords:** hassle costs; charging model; platform operation strategy; network effect

## 1. Introduction

In the digital age, platforms are becoming more and more prevalent as they offer a variety of integrated services to consumers, becoming the mainstream and direction of resource allocation [1]. Traditional marketing believes that "more is better" and that a variety of products will attract consumers and increase market share. However, research in the field of consumer behavior has found that when the number of products rises to a certain level, consumers experience a choice overload effect, delaying or even not making a decision, and experiencing reduced satisfaction and increased regret [2]. However, the occurrence of choice overload is related to consumer preferences, and there are generally two different types of consumers, those who are willing to spend more time selecting goods in search of lower prices when making transactions, and those who want to select the goods they need in the shortest possible time. Hviid and Shaffer claim that consumers must spend time on price searches, filling out forms, and shipping to meet the restrictions attached to these minimum price matches; these are called hassle costs [3]. When the number of merchants on a platform is high, consumers incur higher hassle costs when transacting and may trigger a choice overload effect for consumers. Yanhong Sun [4] found that when consumer sensitivity to the hassle cost is relatively high, offering web showrooms in the duopoly market may increase the offline retailer's profit but reduce the online retailer's profit, which is counterintuitive. Ruixiao Kong et al. [5] believe that BOPS is not always beneficial to the retailer, depending on the unit operating cost and customer hassle costs in the BOPS channel and the cross-selling profit. Fei Gao and Xuanming Su [6] built a stylized

model where a retailer operates both online and offline channels by providing real-time information about inventory availability and by reducing the hassle costs of shopping. Jiajia Cao [7] believes that the firm should enter the platform if the annual service fee is relatively low, otherwise, the firm should not enter the platform. Interestingly, in the case of a firm with an offline store with relatively large operational or hassle costs, the firm is more reluctant to enter the platform. Qian Zheng and Manman Wang [8] examine the conditions under which the supply chain members should cooperate to adopt the deliver-from-store model and further investigate the impact of consumers' freshness sensitivity and offline hassle costs on supply chain members' sales model options. The current study does not take into account the impact of the different hassle costs incurred by consumers due to the different number of merchants in a transaction on the platform operation strategy. As bilateral users in the platform market are interdependent, usually platform enterprises will select bilateral users by setting platform transaction rules and reviewing platform users' qualifications, so as to establish their positioning in a certain market segment to provide more specialized and refined services, forming differentiated competition and creating a competitive advantage for platform enterprises, which will not only enrich market choices, but also enhance the competitive intensity [9].

In a bilateral market, user scale is the foundation of a healthy platform ecosystem and ensures that the platform can grow. The fee system of the platform for bilateral users affects both the size of bilateral users and directly affects the platform revenue. Currently, the main forms of fees charged by platforms to merchants can be categorized as registration fee, transaction fee, and two-part fees [10]. For example, e-commerce platform companies such as Tmall and Jingdong adopt a two-part fees system, charging merchants an annual fee for their services and also taking a certain commission from the final transaction amount based on the services provided to merchants; comparatively, Taobao adopts a transaction fee system, where merchants are not charged for opening a shop but have to pay a commission based on the transaction amount when using Taobao's services. For merchants, registration fees can be seen as a barrier to entry and will limit the number of merchants, but because transaction costs are lower thereafter, they can attract high-volume merchants or provide incentives for merchants to increase transaction volumes, to the detriment of the platform; if a transaction fee system is adopted for merchants, the setting of a transaction fee percentage is critical, as a lower fee may result in an influx of merchants, affecting the cost of hassle for consumers, while a higher fee may affect merchants' revenue and make the platform unattractive. For example, under the competitive pressure of Pinduoduo, the current Taobao platform has a tendency to shift the merchant population. Under the assumption of undifferentiated users, Armstrong [11] finds that a registration fee system with discriminatory pricing for bilateral users is effective in facilitating the formation and development of bilateral markets in the early stages of platform development. Daozhi Zhao and Yang Xue [12] provide references for decisions of the manufacturer and surplus production capacity supplier to join in sharing, as well as corresponding optimal pricing strategies, which guide platforms to keep a balance between profitability and attracting participants by relatively low access requirements and commission rates. Chen et al. [13] study the equilibrium pricing and subsidy strategies of bike-sharing platform firms when customers have time-sensitive preferences. Taylor [14] studies the effects of customer time sensitivity and seller agency independence on the optimal price of on-demand service platforms. De-jian Xia and Yong Wang [10] believe that when there is an equilibrium in the choice of charging system for online merchants by the two oligopolistic platforms, it must be a separation equilibrium in which one party adopts a transaction-based charging system and the other party adopts a two-part charging system. The current research on merchant pricing models only considers the optimal pricing of platforms under different situations and the impact of a single pricing model on the optimal decision of platforms.

It can be seen that the current research only considers the impact of consumer behavior on platform pricing and channel selection, and the impact of different pricing models on the platform's decision in terms of platform charging models, without considering

the impact of different charging models on the platform's operation strategy under the hassle costs incurred by consumers in transactions. Therefore, this paper will consider how to charge merchants in the platform operation strategy under different hassle costs for consumers and analyze the influencing factors. In summary, given the network externalities, non-neutral price structure, interdependence, and complementarity of the platform economy, it is necessary to consider the impact of differences in consumer preferences when formulating platform operating strategies. Therefore, this paper discusses two types of platform operation strategies to address consumers' preference for hassle costs; one is to meet consumers' demand for diversity by offering a wide selection of products, such as Android and Walmart, etc., and this is called "diversified platform", while the other is to reduce the cost of hassle and enhance the shopping experience of customers through the careful selection of merchants' products, such as Apple's IOS platform and Costco supermarkets, referred to as "selected platform". The article first establishes a basic benefit model for consumers to join diversified platforms and selected platforms according to consumers' different preferences for hassle costs, and establishes a basic benefit model for merchants according to the traditional registration fee model, and analyzes this bilateral benefit model. The aim is to determine the bilateral user scale and profit at equilibrium, and compare and analyze the operation strategy of diversified platforms and selected platforms to obtain which operation strategy the platform should choose. After the optimal operation strategy is obtained, according to the different charging modes (registration fee system and transaction fee system) that the platform can adopt to merchants, a benefit function is established, and a balanced analysis is conducted to finally determine the charging mode that the platform should adopt to merchants. Finally, according to the equilibrium user scale, and pricing and profit under the optimal operation mode, the influence of the main influencing factors is analyzed, and suggestions are given for the development of the platform's operation strategy, providing a theoretical basis for the platform's decision-making.

The paper is structured as follows. Section 2 provides an overview of the background literature, Section 3 outlines the basic model, Section 4 analyses the platform operating strategy, Section 5 analyses the main influencing factors of the platform strategy, Section 6 details the analysis of the results, and Section 7 provides concluding remarks and management insights.

## 2. Literature Review

### 2.1. Research on Consumer Buying Behavior

Current research on aspects of consumer user behavior has focused on consumer repetitive purchase intentions [15–19], consumer preference differences [20–25], consumer hassle costs [26–32], and choice overload effects. Traditionally, economics [33], psychology [34], and marketing [35] have argued that having more choices is better than having fewer choices, however, in complex purchasing environments, users experience negative effects in determining the value of a service in order to find more alternatives [36,37], such as regret [38] and dissatisfaction [39,40]. Hviid and Shaffer et al. found that consumers incur hassle costs in price search when they request a refund [3]. Subhasish Dugar et al. experimentally studied price matching guarantees and found that the presence of arbitrary positive hassle costs may completely undermine buyer motivation [41]. Anthony Dukes et al. studied hierarchical customer service organizations with a hierarchical structure and found that these structures may frustrate customers seeking a high level of remediation, and that this dissatisfaction imposes hassle costs on customers, so they seek lower levels of redress [42]. Nagar et al. [43], by investigating the extent to which the level of diversity consumers seek when shopping online can overload the myriad of choices they are offered, found that too much choice in online shopping can lead to a decrease in consumers' willingness to patronize merchants. A larger number of merchants within a bilateral platform leads to a choice overload effect, which in turn leads to larger hassle costs for consumers when transacting. Fogel and Thornton found that too much hassle discourages consumers from using rebates, while too little hassle induces all customers to use rebates [27]. Popkowski

et al. examined the effect of hassle costs on optimal rebate strategies and showed that the hassle of rebates level plays a key role in dividing the market into rebate and non-rebate purchasers [44].

The analysis shows that current research on consumer purchase behavior has focused on purchase preferences, choice overload, the impact of hassle costs on purchase intentions, and the impact of purchase intentions on platform management decisions, without considering the impact of hassle costs on platform operation strategies within the platform due to the hassle costs incurred when consumers transact.

*2.2. Research on the Charging Mode of Platform to Merchants*

Current research on platform charging models for merchants focuses on a single strategy, considering only which charging model is optimal when consumers are homogeneous, the impact of the number of transactions and users' psychological costs on the charging model, and the discriminatory pricing strategy based on consumers' purchasing behavior. The current fee systems of platforms for online merchants include three fee systems: registration fee system, transaction fee system, and two-part fee system [10,45,46]. Armstrong [47] analyzed the fee models of monopolistic platforms when charging registration fee, transaction fee, and two-part fee, and found that network externalities are the main reason for the existence of asymmetric pricing on platforms. Rochet and Tirole [45] and Caillaud and Jullien [48] found that the demand of users on one side will heavily depend on the size of users on the other side when the platform charges transaction fees to both users. Rysman [49], Zingal and Becker [50], Shi et al. [51], and Zhu et al. [52] in a two-part fee system found that the platform would set low prices for users on one side and high prices for users on the other side, which reflects the pricing characteristics of a non-neutral price structure. Hagiu [53] discovered that the optimal pricing for bilateral users under customer diversity preferences is the platform charged registration fee. Xinyu Pan et al. [54] used the hoteling model to solve the optimal pricing strategy when maximizing the profit of the cloud manufacturing service platform, and derived the optimal allocation result of the platform's production capacity. Dawei Liu [55] introduced the category of users' psychological cost into the decision analysis of platform firms' charging models, and analyzed the impact of the change of psychological cost level on the choice of platform charging models. He found that for the dual oligopolistic competitors, the final equilibrium is that one platform adopts the transaction fee model and the other platform adopts the registration fee model in both static and dynamic games. Fan et al. [56] analyzed the effects of different fee strategies by e-retailers under the introduction of the autonomous sales model and found that the autonomous sales model is more likely to significantly increase product demand when the platform charges proportionally to sales compared to the platform charging by volume. Zou Jia [57] analyzed the application strategies of four pricing models in bilateral markets: registration fee, transaction fee, two-part fee and profit sharing system, and found that although transaction fee or two-part fee is more beneficial for the platform to obtain a higher percentage of surplus from sellers, it does not necessarily lead to higher profits for the platform because it distorts the effort level of sellers. Some scholars have studied discriminatory pricing strategies based on consumer purchase history (BBPD) [58–65], where Kai Zhang [66] studied consumers' stay and transfer behavior in the model by assuming that consumers have different user experiences and found that firms always set high prices for regular consumers and low prices for new consumers. Jing B [64] and Jain S [65] investigated how consumer preferences affect the BBPD.

Thus, it can be seen that the current studies on the platform's charging model for merchants focus on the optimal charging model choice for homogeneous consumers, the influencing factors of the charging model, and the BBPD behavior, but there is limited literature on the platform's charging model for merchants based on consumers' purchasing propensity to maximize the platform's profit.

In summary, the charging model of the platform affects the number of merchants in the platform, and the excessive number of merchants will generate a selection overload

effect, which will lead to greater hassle costs for consumers when trading, affect consumers' willingness to purchase, and ultimately affect the platform management decisions. Therefore, this paper mainly considers the charging models of the platform for merchants under the different hassle costs incurred by consumers in transactions, and provides a theoretical basis for the new platform to formulate management strategies.

## 3. Establishment of the Basic Model

This paper studies the market operation strategy of a two-sided platform consisting of merchants and consumers. It is assumed that users on both sides of the platform are single-owned and consumers are sensitive to hassle costs. Taking into account the differences in consumers' perception of hassle costs in the market, the platform can formulate two operation strategies: diversified platforms and selected platforms. Among the diversified platforms, the platform improves the diversification of platform services by introducing a large number of merchants. The platform usually provides fewer basic services for the transaction process of both parties, and consumers need to pay higher hassle costs and spend more time to complete commodity transactions. Among the selected platforms, the platform usually improves the access standards of merchants to select a small number of high-quality merchants to settle in, and provides more basic services for the transaction process of both parties to improve transaction efficiency. Consumers can pay lower hassle costs to complete commodity transactions faster. Therefore, the benefits consumers gain from the platform include the basic services obtained from the platform, the interactive utility obtained from the cross-network effect, the hassle costs of transactions, and the cost of obtaining value. Merchant's utility comes from interaction benefits from cross-network effects, revenue from platform services, operating expenses, and fees paid to the platform.

Let $i$ denote the platform number, $i \in \{1,2\}$, where 1 and 2 represent diversified platforms and selected platforms, respectively. In this paper, $\alpha_i$ is used to represent the marginal revenue brought to consumers by each additional merchant in the platform, and $\beta_i$ is used to represent the marginal revenue brought to the merchant by each additional consumer in the platform. Let $k$ denote the hassle costs in diversified platforms and $\gamma k$ denote the hassle cost in selected platforms, and $0 < \gamma, \eta, \alpha_1, \beta_1, \alpha_2, \beta_2 < 1$. Due to the heterogeneity in the perceived utility of the platform by consumers within the platform, let $\tau$ be a random variable of the consumer's utility perception heterogeneity; $f$ is the degree of heterogeneity of the fixed cost of the merchants to provide products or services on the platform, and $\tau$ and $f$ are uniformly distributed on (0, 1), so the following model can be constructed:

The utility function obtained by consumers joining diversified platforms is:

$$U_{c1} = V_{h1} + \tau + (\alpha_1 - k)N_{s1} - P_{h1} \tag{1}$$

The utility function consumers gain when they join the selected platforms is:

$$U_{c2} = V_{h2} + \tau + (\alpha_2 - \gamma k)N_{s2} - P_{h2} \tag{2}$$

The utility function obtained by merchants joining the platforms is:

$$U_{s1,2} = \beta_i N_{ci} + V_{si} - P_{si} - f \tag{3}$$

In the formula: $U_{c1}$, $U_{c2}$ represent the utility obtained by consumers in diversified platforms and selected platforms, respectively, $U_{s1}$, $U_{s2}$ represent the utility obtained by merchants joining diversified platforms and selected platforms, respectively, and $P_{hi}$ represents the registration fee paid by consumers to join platform $i$, $V_{si}$ represents the revenue that the merchant obtains from the basic services of platform $i$, and $P_{si}$ represents the registration fee charged by the platform $i$ to the merchant. $N_{si}$ represents the number of merchants in platform $i$, and $N_{ci}$ represents the number of consumers in platform $i$. $V_{hi}$

represents the revenue that consumers in platform $i$ obtain from platform services, and $0 < V_{hi}$, $V_{si1} < 1$.

When the user joins the platform's utility $U_{c1}$, $U_{c2}$, $U_{s1}$, $U_{s2} > 0$, the user will join the platform. Therefore, in order to increase the scale of users and profits, the platform should formulate the charging mode for merchants according to the purchasing tendency of consumers, and choose the best development mode of the platform to ensure the maximization of platform profits under a balanced user scale.

## 4. Platform Operation Strategy Analysis

### 4.1. Determine User Scale and Profit at Equilibrium

Users usually choose the platform they join based on the utility and whether their utility of joining the platform is greater than zero, i.e., whether $U_{c1}$, $U_{c2}$, $U_{s1}$, $U_{s2}$ is greater than zero. Therefore, we assume that there exists a marginal amount of users $f_1$, $\tau_1$, when $f \leq f_1$, $\tau \geq \tau_1$, users will choose to join the platform only when the benefits they gain in the platform $U_{ci} > 0$, $U_{si} > 0$. The equilibrium states of the diversified and selected platforms are discussed next.

#### 4.1.1. Diversified Platform

When a user joins a diversified platform, $f \leq f_1$, $\tau \geq \tau_1$, the user utility $U_{c1}$, $U_{s1}$ are greater than zero. At this time, the user scale of buyers and sellers is: $N_{c1} = 1 - \tau_1$, $N_{s1} = f_1$. i.e.,

$$N_{c1} = 1 - \tau_1 = 1 + (V_{h1} - P_{h1}) + (\alpha_1 - k)N_{s1} \tag{4}$$

$$N_{s1} = f_1 = \beta_1 N_{c1} + V_{s1} - P_{s1} \tag{5}$$

$N_{c1} = N_{c1}^*$, $N_{s1} = N_{s1}^*$ in equilibrium, so the pricing level in equilibrium needs to satisfy:

$$P_{h1} = 1 + V_{h1} + (\alpha_1 - k)N_{s1}^* - N_{c1}^* \tag{6}$$

$$P_{s1} = \beta_1 N_{c1}^* + V_{s1} - N_{s1}^* \tag{7}$$

The expression for the equilibrium profit function of the platform is:

$$\pi_1 = P_{h1}N_{c1}^* + P_{s1}N_{s1}^*$$

Taking Equations (6) and (7) into the profit expression yields:

$$\pi_1 = [1 + V_{h1} + (\alpha_1 - k)N_{s1}^* - N_{c1}^*]N_{c1}^* + [\beta_1 N_{c1}^* + V_{s1} - N_{s1}^*]N_{s1}^* \tag{8}$$

From Equation (8), the Hesse matrix is obtained as:

$$\begin{bmatrix} -2 & \alpha_1 - k + \beta_1 \\ \alpha_1 - k + \beta_1 & -2 \end{bmatrix}$$

From the Hesse matrix, the sequential master-sub type $M_1 = -2 < 0$, $M_2 = 4 - (\alpha_1 - k + \beta_1)^2$. We assume that $M_2 > 0$ to obtain a negative definite Hesse matrix, the profit function has a maximum value.

Find the first order derivative of the equilibrium user size $N_{c1}^*$, $N_{s1}^*$ in the expression of the profit function such that $\frac{\partial \pi_1}{\partial N_{c1}^*} = \frac{\partial \pi_1}{\partial N_{s1}^*} = 0$. The user scale at equilibrium can be obtained as follows:

$$N_{c1}^* = \frac{(\alpha_1 - k + \beta_1)V_{s1} + 2(1 + V_{h1})}{4 - (\alpha_1 - k + \beta_1)^2} \tag{9}$$

$$N_{s1}^* = \frac{(\alpha_1 - k + \beta_1)(1 + V_{h1}) + 2V_{s1}}{4 - (\alpha_1 - k + \beta_1)^2} \tag{10}$$

Taking the equilibrium-time user scale Expressions (9) and (10) into the equilibrium pricing, Expressions (6) and (7) yields:

$$P_{h1} = \frac{(1 + V_{h1})(2 + k\beta_1 - \alpha_1\beta_1 - \beta_1^2) + V_{s1}(\alpha_1 - k - \beta_1)}{4 - (\alpha_1 - k + \beta_1)^2} \tag{11}$$

$$P_{s1} = \frac{V_{s1}(2 - \alpha_1^2 - k^2 + k\beta_1 + 2k\alpha_1 - \alpha_1\beta_1) + (1 + V_{h1})(-\alpha_1 + k + \beta_1)}{4 - (\alpha_1 - k + \beta_1)^2} \tag{12}$$

Substituting the user scale and pricing at equilibrium into the optimal profit function, Expression (8) yields:

$$\pi_1^* = \frac{V_{s1}^2 + (1 + V_{h1})(\alpha_1 - k + \beta_1)V_{s1} + (1 + V_{h1})^2}{4 - (\alpha_1 - k + \beta_1)^2} \tag{13}$$

4.1.2. Selected Platform

The derivation process is the same as the previous diversified platform scenario, which can be derived from the user scale, pricing, and profit at equilibrium:

$$N_{c2}^* = \frac{(\alpha_2 - \gamma k + \beta_2)V_{s2} + 2(1 + V_{h2})}{4 - (\alpha_2 - \gamma k + \beta_2)^2} \tag{14}$$

$$N_{s2}^* = \frac{(\alpha_2 - \gamma k + \beta_2)(1 + V_{h2}) + 2V_{s2}}{4 - (\alpha_2 - \gamma k + \beta_2)^2} \tag{15}$$

$$P_{h2} = \frac{(1 + V_{h2})(2 + \gamma k\beta_2 - \alpha_2\beta_2 - \beta_2^2) + V_{s2}(\alpha_2 - \gamma k - \beta_2)}{4 - (\alpha_2 - \gamma k + \beta_2)^2} \tag{16}$$

$$P_{s2} = \frac{V_{s2}(2 - \alpha_2^2 - (\gamma k)^2 + \gamma k\beta_2 + 2\gamma k\alpha_2 - \alpha_2\beta_2) + (1 + V_{h2})(-\alpha_2 + \gamma k + \beta_2)}{4 - (\alpha_2 - \gamma k + \beta_2)^2} \tag{17}$$

$$\pi_2^* = P_{h2}N_{c2}^* + P_{s2}N_{s2}^* = \frac{V_{s2}^2 + (1 + V_{h2})(\alpha_2 - \gamma k + \beta_2)V_{s2} + (1 + V_{h2})^2}{4 - (\alpha_2 - \gamma k + \beta_2)^2} \tag{18}$$

Assuming that the total number of user scale of both parties in the platform is 1, respectively, the larger the user scale of both parties, the larger the share of the user scale of the platform in the total market scale of the platform, i.e., the platform is more competitive in the market.

*4.2. Comparison of Platform Operation Strategies*

Assuming that what the user gains from the cross-network effect is positive, i.e., $\alpha_1 - k + \beta_1 = u_1 > 0$, $\alpha_2 - \gamma k + \beta_2 = u_2 > 0$, because the equilibrium scale of the user and the profit are greater than zero, i.e., $4 - u_1^2 > 0$, $4 - u_2^2 > 0$, so $0 < u_1$, $u_2 < 2$. Let $1 + V_{h1} = m_1$, $1 + V_{h2} = m_2$.

Comparing the bilateral user scale and profit of diversified platforms and selected platforms, we gain:

$$N_{c2}^* - N_{c1}^* = \frac{(u_2 V_{s2} + 2m_2)(4 - u_1^2) - (u_1 V_{s1} + 2m_1)(4 - u_2^2)}{(4 - u_2^2)(4 - u_1^2)} \tag{19}$$

$$N_{s2}^* - N_{s1}^* = \frac{(u_2 m_2 + 2V_{s2})(4 - u_1^2) - (u_1 m_1 + 2V_{s1})(4 - u_2^2)}{(4 - u_2^2)(4 - u_1^2)} \tag{20}$$

$$\pi_2^* - \pi_1^* = \frac{(V_{s2}^2 + u_2 m_2 V_{s2} + m_2^2)(4 - u_1^2) - (V_{s1}^2 + u_1 m_1 V_{s1} + m_1^2)(4 - u_2^2)}{(4 - u_2^2)(4 - u_1^2)} \tag{21}$$

When the strength of the bilateral network effects and the quality of platform services in the two platform operation strategies are the same, i.e., $\alpha_1 = \alpha_2$, $\beta_1 = \beta_2$, $V_{h2} = V_{h1}$, $V_{s1} = V_{s2}$, it is obtained that $N_{c2}^* - N_{c1}^* > 0$, $N_{s2}^* - N_{s1}^* > 0$, $\pi_2^* - \pi_1^* > 0$. Thus, it can be seen that even if the network effect and service quality of the platform remain unchanged, the selected platform operation strategy is better than the diversified platform operation strategy in terms of equilibrium user scale and profit, and in real life, consumers in the diversified platform pay more hassle cost for pursuing the diversification of goods, and the purchasing power of consumers in the selected platform is greater than that in the diversified platform, and there exists $\alpha_1 \leq \alpha_2$, $\beta_1 \leq \beta_2$, so there is $u_2 > u_1$. For example, in the smartphone industry, the IOS platform strictly reviews the APP to ensure quality, while the "Android" platform is less reviewed, so consumers can gain higher revenue from the IOS platform, and so there is $\Delta V \geq 0$, so that $V_{h2} = V_{h1} + \Delta V$, $V_{h2} \geq V_{h1}$. Usually, the selected platform will satisfy customers' needs by attracting fewer and better merchants and high-quality products, while the service objectives of the diversified platform prefer to satisfy consumers' tendency to be more selective, thus, in terms of platform service provision, the shopping process of the selected platform is simpler than that of the diversified platform, and the basic shopping service is more friendly. Thus, in reality, there exists $V_{s2} > V_{s1}$. From the above analysis, we can gain $N_{c2}^* - N_{c1}^* > 0$, $N_{s2}^* - N_{s1}^* > 0$, $\pi_2^* - \pi_1^* > 0$.

It follows that the platform should formulate a selected platform operation strategy. The selected platform operation strategy reduces the cost of hassle when consumers transact, and the utility that consumers obtain from within the platform increases, which leads to the growth of user scale, and the growth of merchant user scale due to cross-network externalities, which ultimately leads to the increase in platform profits.

*4.3. Comparison of Platform Fee Models*

Under the operation strategy of the selected platform, the platform adopts the registration fee system for consumers, and can choose the registration fee system or transaction fee system for merchants. Next, we analyze how the platform should formulate the fee model in the selected platform.

The utility function obtained by merchants joining the selected platform under the registration fee system is:

$$U_{s2} = \beta_2 N_{c2} + V_{s2} - P_{s2} - f \tag{22}$$

The utility function obtained by the merchant joining the selected platform under the transaction fee system is:

$$U_{s22} = (1 - \eta)\beta_2 N_{c22} + V_{s22} - f \tag{23}$$

Among them, $U_{s21}$, $U_{s22}$ denote the utility gained by merchants from joining the platform when the registration fee system and the transaction fee system are adopted for merchants in the selected platform, $V_{s2}$, $V_{s22}$ denote the benefits gained by merchants from the platform basic services when the registration fee system and the transaction fee system are adopted for merchants in the selected platform.

The equilibrium user scale, pricing, and profit when adopting the transaction fee system for merchants in the selected platform can be derived as follows:

$$N_{c22}^* = \frac{(\alpha_2 - \gamma k + \beta_2)V_{s22} + 2(1 + V_{h2})}{4 - (\alpha_2 - \gamma k + \beta_2)^2} \tag{24}$$

$$N_{s22}^* = \frac{(\alpha_2 - \gamma k + \beta_2)(1 + V_{h2}) + 2V_{s22}}{4 - (\alpha_2 - \gamma k + \beta_2)^2} \tag{25}$$

$$P_{h22} = \frac{(1 + V_{h2})(2 + \gamma k \beta_2 - \alpha_2 \beta_2 - \beta_2^2) + V_{s22}(\alpha_2 - \gamma k + \beta_2)}{4 - (\alpha_2 - \gamma k + \beta_2)^2} \tag{26}$$

$$\eta = \frac{\left[2 - (\alpha_2 - \gamma k)^2 + \beta_2(\gamma k - \alpha_2)\right]V_{s22} - (1 + V_{h2})(\alpha_2 - \gamma k - \beta_2)}{\beta_2[(\alpha_2 - \gamma k + \beta_2)V_{s22} + 2(1 + V_{h2})]} \tag{27}$$

$$\pi_{22}^* = P_{h22}N_{c22}^* + \eta_2\beta_2 N_{c22}^* N_{s22}^* = \frac{V_{s22}^2 + (1 + V_{h2})(\alpha_2 - \gamma k + \beta_2)V_{s22} + (1 + V_{h2})^2}{4 - (\alpha_2 - \gamma k + \beta_2)^2} \tag{28}$$

When a platform adopts a transaction fee system, it is in the platform's interest to improve the transaction process to increase merchants' transaction volume, and thus there is an incentive to improve value-added services to merchants. Thus, it can be argued that under the transaction fee system, the platform will provide higher quality platform services in order to promote the number of transactions and increase profits, e.g., the same merchant receives more revenue from the platform services in Tmall than Taobao, so there is $V_{s22} > V_{s2}$. Comparing the registration fee system and transaction fee system for merchants in selected platforms, $N_{c22}^* - N_{c2}^* = \frac{u_2(V_{s22} - V_{s2})}{4 - u_2^2} > 0$, $N_{s22}^* - N_{s2}^* = \frac{2(V_{s22} - V_{s2})}{4 - u_2^2} > 0$, $\pi_{22}^* - \pi_2^* = \frac{V_{s22}^2 - V_{s2}^2 + u_2 m_2(V_{s22} - V_{s2})}{4 - u_2^2} > 0$. Thus, the user scale and profit of the selected platform under the transaction fee system are larger than the registration fee system, so the platform should formulate a selected platform operation strategy and adopt the transaction fee system for merchants. For example, "Jingdong" Mall and "Tmall" Mall both adopt the operation mode of the selected platform and adopt the transaction fee system for merchants, and become the industry leader in the field of e-commerce and continuously attract users to join the platform.

## 5. Analysis of the Influencing Factors of the Selected Platform Strategy of the Transaction Fee System

The analysis in the previous section showed that the combination of selected platform strategies implementing transaction fees outperformed the other strategies in terms of balanced user scale and profitability. Now we will analyze the impact of platform parameters on the effectiveness of this strategy implementation.

### 5.1. The Impact of Hassle Costs

The first-order partial derivatives of $k$ for $N_{c22}^*$, $N_{s22}^*$, $\pi_{22}^*$ are obtained as follows: $\frac{\partial N_{c22}^*}{\partial k} = \frac{-V_{s22}\gamma(4 - u_2^2) - 2u_2\gamma(2m_2 + V_{s22}u_2)}{(4 - u_2^2)^2} < 0$, $\frac{\partial N_{s22}^*}{\partial k} = \frac{-m_2\gamma(4 - u_2^2) - 2u_2\gamma(m_2u_2 + 2V_{s22})}{(4 - u_2^2)^2} < 0$, $\frac{\partial \pi_{22}^*}{\partial k} = \frac{-m_2\gamma V_{s22}(4 - u_2^2) - 2u_2\gamma(V_{s22}^2 + m_2u_2V_{s22} + m_2^2)}{(4 - u_2^2)^2} < 0$. Obviously, the bilateral user scale and profit of the platform both decrease with the increase in hassle costs.

The first-order partial derivative of the equilibrium transaction fee ratio of the platform with respect to the hassle cost $k$ is obtained as:

$$\frac{\partial \eta_2}{\partial k} = \frac{V_{s22}^2\beta_2^2\gamma k^2 + V_{s22}\beta_2\gamma(2V_{s22}\beta_2 + 4m_2)k - \gamma\beta_2\left[V_{s22}^2(\beta_2 - 2) - 2m_2(V_{s22}\beta_2 + m_2)\right]}{\beta_2(u_2V_{s22} + 2m_2)^2} \tag{29}$$

The analysis shows that $\frac{\partial \eta_2}{\partial k} > 0$. Thus, the equilibrium pricing ratio of the platform to the merchant increases with the increase in the hassle costs.

### 5.2. The Impact of Network Effects

Finding the first-order partial derivatives of $\alpha_2$, $\beta_2$ for the equilibrium user scale as well as profit of the transaction fee-based selected platform, we get: $\frac{\partial N_{c22}^*}{\partial \alpha_2} > 0$, $\frac{\partial N_{c22}^*}{\partial \beta_2} > 0$, $\frac{\partial N_{s22}^*}{\partial \alpha_2} > 0$, $\frac{\partial N_{s22}^*}{\partial \beta_2} > 0$, $\frac{\partial \pi_{22}^*}{\partial \alpha_2} > 0$, $\frac{\partial \pi_{22}^*}{\partial \beta_2} > 0$. Therefore, the enhancement of the bilateral network effect can promote the increase in bilateral user scale and profit.

The first-order partial derivatives of $V_{s22}$, $V_{h2}$ for the merchant equilibrium pricing ratio, such that $n = \alpha_2 - \gamma k$, thus: $\frac{\partial \eta_2}{\partial V_{s22}} = \frac{-m_2(n - \beta_2)^2 - 4}{\beta_2(u_2V_{s22} + 2m_2)^2} > 0$, $\frac{\partial \eta_2}{\partial V_{h2}} = \frac{-V_{s22}(4 - u_2^2)}{\beta_2(u_2V_{s22} + 2m_2)^2} < 0$, showing that the platform's equilibrium pricing ratio for merchants increases with $V_{s22}$ and

decreases with $V_{h2}$. Next, we analyze the effect of the bilateral network effect characteristics on the equilibrium transaction fee ratio of the platform for a given platform service quality.

The first-order partial derivatives of $\alpha_2$, $\beta_2$ for the merchant equilibrium pricing ratio are obtained as follows:

$$\frac{\partial \eta_2}{\partial \alpha_2} = -\frac{2m_2^2 + 4u_2 m_2 V_{s22} + \left(2 + u_2^2\right) V_{s22}^2}{\beta_2 (2m_2 + u_2 V_{s22})^2} \tag{30}$$

$$\frac{\partial \eta_2}{\partial \beta_2} = \frac{a\beta_2^2 + b\beta_2 + c}{\beta_2 (u_2 V_{s22} + 2m_2)^2} \tag{31}$$

where $a = -(m_2 - nV_{s22})V_{s22}, b = 2V_{s22}(n^2 V_{s22} - 2V_{s22} + m_2 n), c = (n^2 V_{s22} - 2V_{s22} + m_2 n)(2m_2 + nV_{s22})$.

Obviously, $\frac{\partial \eta_2}{\partial \alpha_2} < 0$, showing that the equilibrium pricing ratio of the platform to merchants decreases with the enhancement of $\alpha_2$, and at the same time, because $\frac{\partial \eta_2}{\partial n} < 0$, i.e., the equilibrium pricing ratio of the platform to merchants decreases with the enhancement of the profitability of consumers $n$. When the profitability of consumers is stronger, the scale of users increases, and the benefits of merchants increase due to cross-network externalities, so the platform should reduce the pricing ratio to merchants as a way to encourage merchants to move in.

The sign of $\frac{\partial \eta_2}{\partial \beta_2}$ is determined by the value of the function $f(\beta_2) = a\beta_2^2 + b\beta_2 + c$. Because $a = -(m_2 - nV_{s22})V_{s22} < 0$, thus $f(\beta_2)$ is a quadratic curve with downward opening and the axis of symmetry $\frac{-b}{2a} = \frac{n^2 V_{s22} - 2V_{s22} + m_2 n}{m_2 - nV_{s22}} = \frac{\left(n + \frac{m_2}{V_{s22}}\right)^2 - 2 - \frac{m_2^2}{4V_{s22}^2}}{\frac{m_2}{V_{s22}} - n}$, when $\left(n + \frac{m_2}{V_{s22}}\right)^2 - 2 - \frac{m_2^2}{4V_{s22}^2} = 0$, we gain $n_0 = \sqrt{2 + \frac{m_2^2}{4V_{s22}^2}} - \frac{m_2}{V_{s22}}$

(1) When $0 < n < n_0$, at this time $\frac{m_2}{V_{s22}} < 1.63$, given $\alpha_2$ case at this time the cost of hassle is larger. The symmetry axis is less than zero, then $b < 0$, $c < 0$. At this time, the discriminant of the root $b^2 - 4ac > 0$, the equation $f(\beta_2) = 0$ has two real roots, and both are greater than zero, contradicting that the symmetry axis is less than zero, so there is no such case; when $b^2 < 4ac$, the equation $f(\beta_2) = 0$ has no real roots, so $\frac{\partial \eta_2}{\partial \beta_2} < 0$. The condition of $\frac{m_2}{V_{s22}} < 1.63$ requires $V_{s22}$ to have a larger value, so in this case, the platform provides more value-added services to the merchant to improve the merchant's revenue, and if the merchants benefit more from consumers at this time, the platform will charge the merchants a lower proportion.

(2) When $n > n_0$, the axis of symmetry is greater than zero, when the cost of hassle is smaller for a given $\alpha_2$ case. $b > 0$, $c > 0$, $0 < \frac{-b}{2a} < 1$, when $b^2 - 4ac > 0$, the equation $f(\beta_2) = 0$ has one positive root and one negative root. Let $B = -b - \sqrt{b^2 - 4ac} - 2a$. When $B > 0$, the equation $f(\beta_2) = 0$ has a real root x at $\beta_2[0,1]$, when $\frac{\partial \eta_2}{\partial \beta_2}$ decreases monotonically on $\beta_2[0,1]$ and crosses the horizontal axis, thus there is as $\beta_2$ increases the pricing ratio $\eta_2$ which then increases and then decreases. When $B \leq 0$, $\frac{\partial \eta_2}{\partial \beta_2}$ is positive but gradually decreasing in $\beta_2 \in [0,1]$, the equilibrium pricing ratio of the platform to merchants increases with the enhancement of the merchant network effect. The first-order partial derivatives of $\gamma k$ and $\alpha_2$ in $B$ yield that $\frac{\partial B}{\partial \gamma k} > 0$, $\frac{\partial B}{\partial \alpha_2} < 0$, i.e., $B$ increases with the increase in $\gamma k$ and decreases with the increase in $\alpha_2$.

## 6. Results and Simulation Analysis

### 6.1. The Choice of Platform Strategy

According to the analysis in Section 4, numerical simulations of the three strategies are performed using MATLAB R2018a without loss of generality, and the specific parameters can be set as $\alpha_1 = 0.2$, $\alpha_2 = 0.3$, $\beta_1 = 0.2$, $\beta_2 = 0.3$, $V_{h1} = 0.2$, $V_{h2} = 0.3$, $V_{s1} = 0.15$, $V_{s2} = 0.2$, $V_{s22} = 0.25$, $\gamma = 0.1$, and the results are shown in Figure 1.



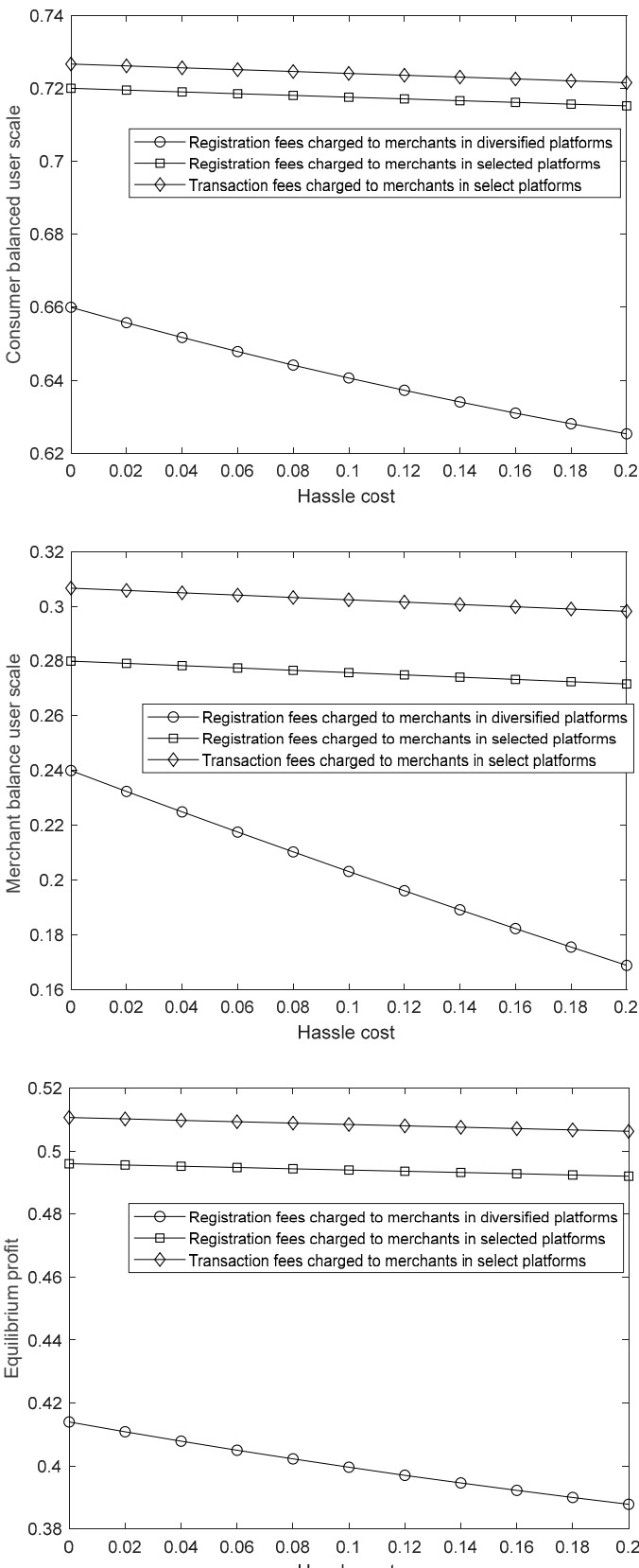

**Figure 1.** Impact of hassle costs on bilateral user size and profit.

Therefore, the platform should adopt a selected platform operation model and adopt a transaction fee charging model for merchants to promote maximum platform profits.

As the cost of hassle increases, the benefits to the consumer decrease, so the platform should choose fewer and better merchants and high-quality products to attract customers. Smartphone companies, such as Apple, strictly control the quality as well as the quantity of merchants to create a good shopping environment for consumers, and adopt a transaction fee system for merchants. Apple's App Store will generate a total revenue of more than USD 64 billion in 2020, which is higher than USD 50 billion in 2019 and USD 48.5 billion in 2018, and the profit is up.

*6.2. The Impact of Consumer Hassle Costs*

The bilateral user scale and profit of the platform are decreasing with the increase in hassle costs. Therefore, when adopting the selected platform strategy, the platform should make efforts to reduce the cost of hassle for consumers, including streamlining the number of homogeneous merchants in the platform, improving the transaction process, optimizing the interaction interface, and providing transaction services and other means to increase the incremental value of consumers in each transaction, thus promoting the growth of bilateral user scale and revenue. For example, Jingdong Mall adopted a selected platform operation model in the early stage of development, strictly controlling the quality and quantity of merchants, improving the quality and reducing the quantity of products, providing consumers with a good shopping environment, improving consumer satisfaction and successfully attracting and retaining a large number of consumers on the platform. Smartphone companies such as Apple have also adopted a selected platform operation model, with profits reaching 83.4% of total smartphone industry profits in 2017 when shipments were only 14.7%. Costco, a mass-market warehouse membership store, has become a retail leader with fewer and better SKUs, with a membership scale of over 96 million and revenue size of USD 141.6 billion at the end of 2018.

The platform's equilibrium pricing ratio for merchants increases as the cost of hassle increases. When consumers incur larger hassle costs due to larger merchant user scale, consumers receive fewer benefits from the platform, leading to a decrease in consumer user scale, so the platform should increase the percentage of fees charged to merchants to control the number of merchants entering the platform while increasing the total platform profit.

*6.3. The Impact of Network Effects*

From the analysis in Section 5.2, it is clear that the enhancement of the bilateral network effect can promote the increase in bilateral user scale and profit. The platform's service quality $V_{s22}$ and $V_{h2}$ to bilateral users have a key impact on the proportion of fees charged by the platform to merchants. When a platform service is given, as the consumer side network effect in the platform increases and the hassle costs decreases, the pricing ratio of the platform to merchant decreases, but the specific trend is influenced by the strength of the bilateral network effects.

When adopting the transaction fee charging system for merchants, the platform should set the appropriate fee ratio according to the intensity of the network effects of bilateral users in the platform, the cost of hassle and the quality of platform services. For example, both Tmall and Jingdong Mall adopt the transaction fee charging system for merchants, but due to the inconsistent characteristics of the platforms, the two platforms set different charging standards for merchants.

Using MATLAB R2018a to simulate the equilibrium transaction fee ratio of platforms numerically, the specific parameters can be set as $V_{h2} = 0.5$, $V_{s22} = 0.2$, Figure 2a $\gamma = 0.3$, $k = 0.3$, $\alpha_2 = 0.3$, Figure 2b $\gamma = 0.1$, $k = 0.3$, $\alpha_2 = 0.3$, Figure 2c $\gamma = 0.1$, $k = 0.3$, $\alpha_2 = 0.4$, without loss of generality, and the results are shown in Figure 2. From the figure, it can be seen that under the general trend, the proportion of the platform equilibrium transaction fee gradually decreases with the enhancement of the merchant network effect, and the bilateral equilibrium user scale and profit is the best in Figure 2c, i.e., the platform will achieve a larger market share and profit under the situation of smaller hassle costs and larger consumer network effect.

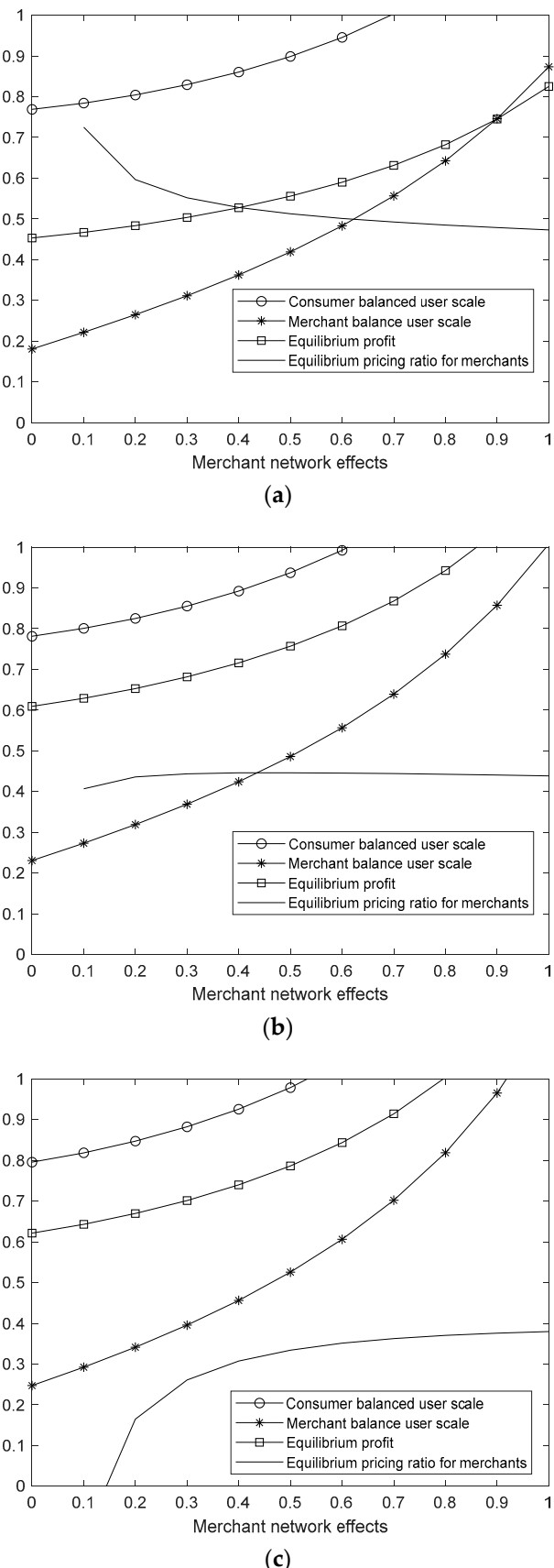

**Figure 2.** Effect of merchant network effects on the proportion of equilibrium transaction fees on the platform under different conditions. (**a**, **b**, **c** are the equilibrium results for different parameters).

## 7. Conclusions and Management Insights

The scale of platform users is an important basis for platform growth, and platform profit is the primary concern of platform managers. Therefore, this paper considers how the new platform should formulate the development model from the aspect of bilateral users, i.e., how to formulate the fee model of merchants under the different hassle costs arising from the number of merchants in consumer transactions, and establishes the utility function and profit function of bilateral users on this basis to produce the following conclusions and management insights:

### 7.1. Platform Operation Strategy

The new platform should formulate the operation model of the selected platform and adopt the transaction fee charging system for merchants. The number of merchants in the platform is small and precise under the operation mode of the selected platform, so that consumers can choose the right goods for themselves faster in the transaction, the cost of hassle incurred by consumers in the transaction is smaller, and the benefits of joining the platform are greater, which results in there being more consumers joining the platform, the scale of the consumer increases, and the number of merchants increases due to cross-network externalities. The adoption of a transaction fee-based fee model for merchants will result in increased benefits for merchants and increased scale of the merchant, ultimately leading to increased profits for the platform. Therefore, the platform should control the quality and quantity of merchants, reduce the cost of hassle for consumers, create a better trading environment for consumers, improve consumer satisfaction, and promote platform transactions.

### 7.2. The Impact of Consumer Hassle Costs on Transaction Fee-Based Selected Platform Strategies

In the transaction fee system, selected platform strategy, bilateral user scale, and platform profits decrease with the increase in hassle costs, the increase in hassle costs leads to a decrease in the benefits of consumers joining the platform, which in turn leads to a decrease in the scale of consumer users, and a decrease in the scale of merchant users due to the existence of cross-network externalities, which in turn leads to a decrease in platform profits. The equilibrium pricing ratio of the platform to the merchant increases with the increase in the hassle costs. When the hassle costs of the consumer transaction increase, the benefit of the consumer decreases, leading to the decrease in the consumer user scale, and the platform gains less revenue from the consumer and needs to increase the fee ratio to the merchant to increase the overall revenue of the platform. Increasing the number of merchants will bring larger hassle costs, and the platform should therefore increase the percentage of charges to merchants.

### 7.3. The Impact of Bilateral Network Effects on Transaction Fee-Based Selected Platform Strategies

The service quality of the platform to bilateral users and the network effects of bilateral users jointly determine the trend of the platform's fee percentage to merchants. In general, the less value-added services for consumers, the more value-added services for merchants, and the smaller network effect on the consumer side, the higher the percentage of charges to merchants. The smaller the platform's value-added services to consumers or the smaller the consumer network effect, the smaller the consumer benefits, which in turn affects the revenue obtained by the platform from consumers, and the platform will therefore increase the pricing ratio for merchants to improve the overall revenue. the platform will increase the value-added services for merchants to increase the proportion of their fees, but under the joint effect of these three factors, the percentage of charges to merchants by the platform does not show a monotonous trend of change, but three different dynamics of change However, the general trend is that the greater the network effect on the merchant's side, the lower the fee ratio and the higher the platform's profit. When the merchant network effect increases, the platform is more attractive to merchants, and at this time the platform will reduce the percentage of fees charged to merchants in order to obtain more merchant user

scale, which ultimately maximizes the profit of the platform. Therefore, the platform should choose the transaction fee system to select the platform strategy and integrate the quality of service, the cost of hassle, and the strength of the network effect to set the fee ratio.

### 7.4. Research Limitations and Recommendations

(1) This paper assumes a single attribution of bilateral users within the platform and does not take into account the existence of the multi-attribution of users, which needs to be reflected in the model in the future to discuss its impact on the development model of the platform.

(2) The dynamic game idea is not considered in the process of model solving, and the model can be derived from the perspective of dynamic game in the future.

**Author Contributions:** Conceptualization, H.L.; Funding acquisition, H.L.; Project administration, H.L.; Supervision, H.L.; Validation, H.L.; Formal analysis, F.L.; Investigation, F.L.; Methodology, F.L.; Resources, F.L.; Writing—original draft, F.L.; Writing—review & editing, F.L.; Data curation, F.D. All authors have read and agreed to the published version of the manuscript.

**Funding:** This research received no external funding.

**Institutional Review Board Statement:** Not applicable.

**Informed Consent Statement:** Not applicable.

**Data Availability Statement:** Not applicable.

**Conflicts of Interest:** The authors declare no conflict of interest.

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
