# Peer review of "Research on Platform Operation Strategy Considering Consumers’ Hassle Costs"

_sustainability, doi:10.3390/su14052634_

Round 1

Reviewer 1 Report

First of all, thank the author(s) for your hard work. I think this is an interesting topic and has pivotal importance in considering consumers’ hassle costs. However, I think the following comments will make it better and more robust.

- The use of abbreviations is not consistent throughout the paper. For instance, BBPD.
- The researcher(s) should pay attention to the research gap that is still not sufficient. Therefore, please add more arguments related to the research gap in the introduction. 
- As far as I have seen, updated references in the introduction part are ignored (Hviid, M.; Shaffer, G. Hassle costs: the Achilles' heel of price-matching guarantees. Journal of Economics & 418 Management Strategy 1999; Armstrong, M. Competition in two-sided markets. The RAND journal of economics 2006, 37, 668 691). So, please develop the introduction part of the paper to include 3 to 6 latest references (2017 to 2022) from well-known journals in the field and relevant extracts from them. Please try to include 2 to 4 references from Sustainability. 
-Please try to restructure "part 7" Conclusions and Management Insights" in a separate section on research implications. The current writing is not well structured.
-Along the same lines, it is necessary to mention the research limitations and recommendations in a separate section. 
-Finally, please improve the language of the research paper.
I hope that my comments can help you to improve your manuscript.

Author Response

Dear reviewer:

Thank you for your comments concerning our manuscript entitled “Research on Platform Operation Strategy Considering Consumers’ Hassle Costs”. Those comments are valuable and very helpful. We have read through comments carefully and have made corrections.

Based on the instructions provided in your letter, we uploaded the file of the revised manuscript. Revisions in the text are indicated in red font for additions and deletions are indicated in strikethrough font, and add the marker. The responses to the comments are presented following.Please see the attachment for specific changes.

Point 1:The use of abbreviations is not consistent throughout the paper.

Response 1: Standardize abbreviations in the text to make them consistent.(Revisions in the text are indicated in red font for additions and deletions are indicated in strikethrough font,and add the marker)

Point 2:The researcher(s) should pay attention to the research gap that is still not sufficient. Therefore, please add more arguments related to the research gap in the introduction. 

Response 2: Adding more arguments related to the research gap in the introduction. (Revisions in the text are indicated in red font for additions and deletions are indicated in strikethrough font,and add the marker).

Point 3:As far as I have seen, updated references in the introduction part are ignored (Hviid, M.; Shaffer, G. Hassle costs: the Achilles' heel of price-matching guarantees. Journal of Economics & 418 Management Strategy 1999; Armstrong, M. Competition in two-sided markets. The RAND journal of economics 2006, 37, 668 691). So, please develop the introduction part of the paper to include 3 to 6 latest references (2017 to 2022) from well-known journals in the field and relevant extracts from them. Please try to include 2 to 4 references from Sustainability. 

Response 3: Update literature(Revisions in the text are indicated in red font for additions and deletions are indicated in strikethrough font,and add the marker).

Point 4:Please try to restructure "part 7" Conclusions and Management Insights" in a separate section on research implications. The current writing is not well structured.

Response 4: Reconstructing “part 7” (Revisions in the text are indicated in red font for additions and deletions are indicated in strikethrough font,and add the marker).

Point 5:Along the same lines, it is necessary to mention the research limitations and recommendations in a separate section. 

Response 5: Mention the research limitations and recommendations in a separate section(Revisions in the text are indicated in red font for additions and deletions are indicated in strikethrough font,and add the marker).

Point 6:Finally, please improve the language of the research paper.

Response 6: Improve the language in the article(Revisions in the text are indicated in red font for additions and deletions are indicated in strikethrough font,and add the marker).

We would love to thank you for allowing us to resubmit a revised copy of the manuscript and we highly appreciate your consideration.

Sincerely

Ms. Liu

Reviewer 2 Report

1- you have to clarify your research design 

2- need some proofreading for you paper

3- update your references

Author Response

Dear reviewer:

Thank you for your comments concerning our manuscript entitled “Research on Platform Operation Strategy Considering Consumers’ Hassle Costs”. Those comments are valuable and very helpful. We have read through comments carefully and have made corrections.

Based on the instructions provided in your letter, we uploaded the file of the revised manuscript. Revisions in the text are indicated in red font for additions and deletions are indicated in strikethrough font, and add the marker. The responses to the comments are presented following. Please see the attachment for specific changes.

Point 1:you have to clarify your research design.

Response 1: Clarify the research design (Revisions in the text are indicated in red font for additions and deletions are indicated in strikethrough font,and add the marker)

Point 2:need some proofreading for you paper. 

Response 2: Proofreading of papers (Revisions in the text are indicated in red font for additions and deletions are indicated in strikethrough font,and add the marker).

Point 3:update your references

Response 3: Update references (Revisions in the text are indicated in red font for additions and deletions are indicated in strikethrough font,and add the marker).

We would love to thank you for allowing us to resubmit a revised copy of the manuscript and we highly appreciate your consideration.

Sincerely

Ms. Liu

Reviewer 3 Report

Nice work

Author Response

Dear reviewer:

       Thank you very much for your approval!

Sincerely

Ms. Liu